# Combination Therapy of Radiation and Hyperthermia, Focusing on the Synergistic Anti-Cancer Effects and Research Trends

**DOI:** 10.3390/antiox12040924

**Published:** 2023-04-13

**Authors:** Seeun Kwon, Sumin Jung, Seung Ho Baek

**Affiliations:** College of Korean Medicine, Dongguk University, 32 Dongguk-ro, Ilsandong-gu, Goyang-si 10326, Republic of Korea

**Keywords:** radiation, hyperthermia, cancer, combination therapy, synergistic effect, oxygenation, hypoxia, immune response

## Abstract

Despite significant therapeutic advances, the toxicity of conventional therapies remains a major obstacle to their application. Radiation therapy (RT) is an important component of cancer treatment. Therapeutic hyperthermia (HT) can be defined as the local heating of a tumor to 40–44 °C. Both RT and HT have the advantage of being able to induce and regulate oxidative stress. Here, we discuss the effects and mechanisms of RT and HT based on experimental research investigations and summarize the results by separating them into three phases. Phase (1): RT + HT is effective and does not provide clear mechanisms; phase (2): RT + HT induces apoptosis via oxygenation, DNA damage, and cell cycle arrest; phase (3): RT + HT improves immunological responses and activates immune cells. Overall, RT + HT is an effective cancer modality complementary to conventional therapy and stimulates the immune response, which has the potential to improve cancer treatments, including immunotherapy, in the future.

## 1. Introduction

Due to the life expectancy of humans increasing, cancer has become a widespread disease affecting many individuals’ lives. According to statistics from the American Cancer Society, 1,918,030 new cancer cases and 609,360 cancer-related deaths had been reported in the United States by 2022 [1]. Between the mid-1970s and 2011–2017, the five-year relative survival rate for all types of cancers increased from 49% to 68%. Despite significant therapeutic advances, the toxicity of conventional therapies poses a considerable barrier to their practical application [2].

Therapeutic hyperthermia (HT) can be defined as the local heating of a tumor to 40–44 °C, whereas normal tissues are rarely affected by temperatures below 45 °C [3]. During hyperthermia therapy, the temperature of the affected area is raised to around 40–45 °C (104–113 °F) using different methods, such as microwave, radiofrequency, ultrasound, or infrared radiation. Exposing the body or a specific area of the body to high temperatures is conducted to kill cancer cells or make them more sensitive to other cancer treatments, such as chemotherapy or radiation therapy. Because of its greater perfusion and heat transfer ability, normal tissue can resist heat shock better than cancer cells. In addition, HT does not raise the expression of the apoptosis-signaling p53 protein in normal tissue. HT is known to increase the production of reactive oxygen species (ROS), thereby inducing ROS-dependent cellular damage and cancer cell apoptosis [4]. In addition, targeted HT can increase tumor perfusion and blood flow in a temperature- and time-dependent way [5]. Changes in tumor blood flow may enhance vascular permeability, increase oxygenation, decrease interstitial fluid pressure, and help restore physiological pH to normal levels. HT modulates the tumor microenvironment to enhance radiation efficiency [6]. However, because of its toxicity, HT may only be administered to patients in good general health. It is generally considered safe and well-tolerated, but, as with any cancer treatment, it can have side effects. High temperature HT over 45 °C may result in diarrhea, nausea, and vomiting [7]. Heating devices, including ultrasonic, radiofrequency, and microwave devices, must restrict electromagnetic radiation dispersion. HT alone may not be sufficient to destroy tumors when combined with other therapies [8].

Radiation therapy (RT), also known as radiotherapy, remains essential to cancer treatment. Approximately 50% of patients undergo RT [9]. High-energy radiation destroys cancer cells or slows down their growth. The radiation can be delivered externally, using a machine that directs high-energy beams of radiation at the cancer, or internally, by placing radioactive material directly into or near the tumor. As with any cancer treatment, RT can have side effects, which vary depending on the area of the body being treated and the dose of radiation being used. Patients receiving RT have been reported to experience increased discomfort, anxiety, and depression [10]. The primary goal of RT is to inhibit the proliferation of cancer cells. RT is well known to induce oxidative stress, resulting in DNA damage and cell death. RT also influences the tumor microenvironment [11]. Endothelial cells are damaged by radiation, which causes inflammation. Some processes occurring in the tumor microenvironment may paradoxically contribute to the activation of immunosuppression and the induction of radioresistance [12].

Both RT and HT have advantages and disadvantages. According to clinical research, they can be used together to compensate for each other’s shortcomings [13]. When combined, they exert their greatest impact and function in a synergistic manner. In practice, combination therapy may produce a greater rate of cell death at lower HT temperatures, as well as a considerable increase in the local control of cancer, tumor therapeutic signals, and survival rates [7]. Both RT and HT have benefits over anti-cancer medications because they can reach the local target effectively [14]. Accordingly, it is necessary to actively conduct experimental investigations on the efficacy of these two treatments. Based on this, it would be beneficial to devise and use a novel therapeutic strategy in clinical practice. The current research on RT + HT focuses primarily on its clinical efficacy and experimental research on RT + HT is limited. This study reviews research involving the effect of RT + HT in cancer, including experimental approaches of in vitro and in vivo methods and clinical studies. Here, we discuss the effects and mechanisms of RT and HT based on experimental research, as well as prospects for future research.

## 2. Anti-Cancer Effects of RT + HT Treatment

HT acts as a radiosensitizer and makes tumor cells more susceptible to irradiation [15]. When combined with RT, HT can lead to increased cell death in cancer cells, help to shrink the tumor, and improve survival rates [16]. The results are presented in chronological order.

In 1982, Brewer et al. suggested that consecutive RT with HT was an effective therapy for canine fibrosarcoma [17]. Oral fibrosarcomas were heated to 50 °C for 30 s, and nasomaxillary fibrosarcomas were heated to 43 °C for 30 min after RT (32–48 Gy in total). One year after treatment, five of the nine dogs were disease-free.

Campos et al. observed that the body weight gain and survival rate of subcutaneous sarcoma 180-bearing C3H mice were dramatically enhanced by the combination of RT and HT [18]. The percentage of mice with cancer that survived 90 days following therapy with RT, HT, and RT plus HT was 3.33, 16.67, and 76.67, respectively.

TCD50, which measures the number of infectious viruses or bacteria in a sample, was reduced by RT with HT in the FM3A cells of C3H/He mice, as observed by Yamashita [19]. The TCD50 value for RT was 6024 rad, and RT and HT was 5108 rad.

In a study conducted by Dewhirst et al., 43 dogs with primary canine malignant melanomas were treated with either RT alone or RT (8 × 4.6 Gy) combined with HT (43 °C, 20 min) [20]. An increase in temperature led to an increase in the frequency of complete remissions (CR). For tumors treated with RT alone, the CR rate was 21% (3/14). The CR rate increased to 76% (16/21) with HT addition.

Legorreta et al. reported that two months of treatment with a combination of RT (14 × 3.5 Gy) and HT (44 °C, 30 min) successfully controlled a large infiltrating mast cell sarcoma in a dog [21]. The previously untreatable tumor experienced rapid and successful necrosis with combination therapy.

Morphological and morphometric studies on necrosis were conducted by Patrcio et al. to evaluate the therapeutic benefits of HT (43.5 °C, 30 min) combined with RT (2 × 8.5 Gy) [22]. Two tumors were injected into the right lower leg of male BALB/C mice: undifferentiated carcinoma of the mouse breast (Tx) and sarcoma 37 (S37). The level of necrosis induced by the combination treatment was much higher than that induced by a single treatment.

In 1990, Fujiwara et al. designed a study to assess the effects of RT (5 or 10 Gy), HT (46 °C, 60 min), and RT with HT on the morphological changes and vascular permeability of the rabbit VX-2 tumors [23]. At each time point, tumors treated with RT (10 Gy) and HT demonstrated a greater effect on tumor volume and necrosis than those treated with HT or RT alone. When HT was administered following RT (10 Gy), there was a considerable decrease in the tumor volume, and, after 21 days, the tumor appeared to have disappeared.

Ruifrok et al. devised an experiment to investigate the interactions between interstitial RT and interstitial HT (44 °C, 30 min) [24]. Rhabdomyosarcoma type R1 was transplanted into the left flank of the inbred Wag/Rij female rats. The total radiation dose was between 20 and 115 Gy, administered at a rate of 47 cGy/h. After RT and HT, the tumor volume decreased more rapidly than that after RT alone. In addition, a higher cure rate was observed with a lower dose of RT + HT. When treated with RT alone, 80 Gy resulted in several cures, and 115 Gy resulted in a 100% cure rate. In RT + HT cases, tumors treated with 30 Gy produced several cures, and 80 Gy resulted in a 100% cure rate. The cancer cure rate indicates treatment efficacy. The number of animals without tumor growth after treatment is divided by the total number of animals treated.

The thermal enhancement ratio and therapeutic gain factor were determined by Sougawa et al. for combined RT (7.2 Gy/min) and HT treatments of FSa-II tumors in C3Hf/Sed mice [25]. In addition to RT, HT prolonged the tumor growth time. Step-down heating, first at 45.5 °C for 10 min, followed by 41.5 °C for 60 min, resulted in a larger tumor prolongation delay than single heating (45.5 °C, 10 min). Moreover, these treatments alone did not cause any visible thermal injury to the foot, but they reduced the threshold dose shown in the RT dose–response curves for the foot responses.

Nishimura et al. investigated the effect of the timing and sequence of HT (43.5 °C, 45 min) on fractionated RT (5.5–5.6 Gy/min, 86.2–101.7 Gy for five days) of FSa-II tumors in C3Hf/Sed mice [26]. They examined whether HT should be used as an independent agent or used as a RT sensitizer. Regarding the tumor growth time and TCD50 assays, RT combined with HT had a therapeutic advantage over RT alone. The advantage was most evident when HT was combined with RT simultaneously.

Clamping is a method that cuts off the blood supply to a tumor to create complete hypoxia [27]. In 2001, Uma Devi et al. reported that RT (10 Gy) and HT (43 °C, 30 min) affected clamping-induced ischemia and reperfusion in B16F1-melanoma-bearing C57BL mice. Clamping alone or in conjunction with RT had no noticeable effect on apoptosis. HT under clamping induced a greater than 50% increase in apoptotic cells relative to the control and lowered the microvascular density (MVD) to one-third of the control level. Combining clamping with RT and HT increased the number of apoptotic cells to >70% and decreased MVD to one-sixth of the control level. The combination of RT and HT may be advantageous for the treatment of tumors with ischemia-induced acute hypoxia.

In 2002, Ressel et al. found that a combination of RT (5 × 2 Gy), HT (41.8 °C, 60 min), and cisplatin was significantly effective in the treatment of human head and neck squamous cell carcinoma transplanted into mice [28]. RT with HT was more successful than RT or HT alone in terms of the tumor volume and CR. In the RT group (5 × 2 Gy), the number of CR cases (2/15 mice) was the lowest. The combination of HT (41.8 °C) increased the CR rate (5/15 mice).

Rao et al. examined the response of S180, which was developed intradermally in inbred BALB/c mice after treating solid tumors with bleomycin (BLM), RT (10 Gy), and HT (43 °C, 30 min) [29]. The combination of BLM and HT further enhanced the frequency of micronuclei (MN) in the bimodality groups. At 24 h post-treatment, MN counts in the trimodality group (BLM + RT + HT) did not differ significantly from those of the bimodality treatments. However, trimodality-treated tumors exhibited severe tumor necrosis, indicating increased cell loss and immediate tumor regression.

In 2010, Kalthur et al. treated B16F1 melanoma cells grown in adult C57BL mice with Withaferin A (WA) and HT (43 °C, 30 min) to determine whether the RT response was enhanced [30]. Acute RT (30 Gy) and HT increased tumor growth delay compared with RT (40 Gy) alone. Acute RT and HT and WA and RT induced a partial response (PR) of 50% and 62.5%, respectively, indicating a decrease in tumor size of over 30%. However, RT alone did not produce a PR.

Franken applied the linear quadratic model to assess the efficiency of HT (41 °C or 43 °C, 1 h) with low RT (0.5 Gy) doses against high doses (15–30 Gy) [31]. Human SIHA, SW-1573, and RKO cells, as well as rodent V79, R1, and RUC cells, were used. The linear parameter α and quadratic parameter β determine the effectiveness of radiation at low or high doses. An increase in parameter α compared with parameter β indicates that RT and HT are more effective than RT alone. The differences between the values of α and β were larger at 43 °C than at 41 °C.

In 2015, according to Alya et al., partial body HT (PBH) (43 °C, 1 h) prior to γ-radiation (9 Gy) was advantageous in 396 Wistar rats [32]. This helped to restore normal cells following radiation treatment. Gamma-treated rats of both sexes had a lifespan and mortality prolonged by PBH treatment. In addition, PBH improved the recovery of the bone marrow in the femurs and tibias. PBH-treated males had a survival rate of approximately 35% by day 30, whereas all nine Gy-irradiated males died within 16 days.

Borasi et al. theoretically suggested that patients with glioblastoma multiforme (GBM) could be effectively treated with the rapid HT of a focused ultrasound (FUS) device combined with external beam radiation, which exhibits a low level of tumor cell survival and a long offset time [33].

Masunaga et al. discovered the effect of tirapazamine, metformin, or mild-temperature HT (40 °C, 60 min) on RT (2.5 Gy/min)-treated EL4 tumor total cells and quiescent tumor cells [34]. Mild-temperature HT (MTH) after radiation decreased the surviving fraction and increased the frequency of micronuclei and apoptosis compared to radiation alone. Under HT conditions, quiescent cells exhibited comparatively higher micronucleus and apoptosis frequencies than the total cell population.

Marloes et al. evaluated the radiosensitization effect of HT combined with molecular targeting agents, such as PARP1-i, DNA-PKcs-i, and HSP90-i, in cervical cancer cell lines [35]. The non-homologous end-joining or alternative non-homologous end-joining pathway is blocked when HT is coupled with DNA-PKcs-I and PARP1-i, resulting in a more effective radio enhancement.

Rajaee et al. investigated the effects of RT (6 Gy), HT (43 °C, 30 min), and the combination of RT and HT on human prostate carcinoma in cell line DU145 in both monolayer and spheroidal cultures [36]. In both monolayer and spheroid cultures, RT and HT, rather than RT alone, decreased the survival rate as the RT dose increased. RT and HT decreased the survival fraction in both monolayer and spheroid cultures, with greater effects in monolayer cultures.

McDonald et al. used modulated electro-HT (mEHT) (42 °C, 30 min) to evaluate the synergistic effect of RT (5 Gy) on Gs-9L rat gliosarcoma cells, Madin–Darby canine kidney (MDCK) cells, and MCF-7 cells [37]. When RT was combined with mEHT, the percentage of 9 L gliosarcoma cells decreased more than when treated with RT alone, and mEHT enhanced the mortality of 9 L gliosarcoma cells while preserving the survival of non-cancerous cells. On MCF7 or MDCK cells, however, mEHT had no synergistic effects with RT.

Prasad et al. evaluated the increase in the RT dose with HT [38]. The human lung cancer cell lines A549 and NCI-H1299 were used in vitro, and BALB/c nude mice were used in vivo. Adding HT (42 °C) to RT (2.75 Gy/min) increased the cell death and radiosensitivity of the NCI-H1299 and A549 cells. The equivalent RT dose with RT alone was a total of 60 Gy, whereas HT increased the effect of 60 Gy RT to 70.3 Gy for 40.5 °C, 86.3 Gy for 41.6 °C, and 93.6 Gy for 42.4 °C. Compared to RT or HT alone, RT (2 × 5 Gy) plus HT (42 °C, 30 min) significantly decreased the tumor volume and increased the apoptotic cells in an in vivo study.

In 2020, Brüningk et al. examined the response of tumor spheroids created from two human cancer cell lines (HCT116 and CAL27) to single and combination treatments with RT (2 × 5 and 5 × 2 Gy) and HT (47 °C) [39]. Tumor spheroids are three-dimensional structures composed of cancer cells that imitate the shape and organization of normal tissue [40]. Spheroids are frequently utilized in in vitro research to simulate the formation and activity of tumors as they reflect the complexity and variety of actual tumors more accurately than typical two-dimensional cell cultures [41]. The combination of RT and HT shared the same properties with either RT or HT alone, delaying spheroid growth with increasing RT or HT doses [39]. However, RT combined with HT induced the acceleration of spheroid growth compared to treatment with the relevant RT dose. Considering the finding that the size of the spheroid remained less than that of the control within 21 days, it could be inferred that the additional thermal dose further delayed growth when HT was combined with greater RT doses.

Hu et al. reported that non-invasive FUS-induced cavitation sensitized cancer cells to RT and HT [42]. Head and neck cancer (FaDu), glioblastoma (T98G), and prostate cancer (PC-3) cells were subjected to FUS followed by RT (10 Gy) or HT (45 °C, 30 min). FUS-Cav greatly enhanced the susceptibility of cancer cells to RT and HT by reducing long-term clonogenic survival, short-term cell metabolic activity, cell invasion, and activation of sonoporation.

Elming et al. found that combining HT (41.5 °C, 60 min) with low RT was as effective as high RT in inducing the control of C3H mammary carcinomas in CDF1 mice [43]. In addition, researchers have discovered that a shorter gap between RT and HT induces local tumor control. Only RT and cancer with low LET 240 kV resulted in TCD50 to 53 Gy, and 6 MV X-rays resulted in TCD50 to 55 Gy. RT followed by HT resulted in a TCD50 of 44 Gy and 46 Gy, respectively. Table 1 provides a summary of the studies on the effects and mechanisms of HT and RT.

## 3. Physiological Changes Induced by RT + HT

### 3.1. Oxygenation

Rapid and uncontrolled tumor growth restricts the availability of oxygen, leading to hypoxia [44]. The expressions of pro-angiogenic factors, such as vascular endothelial growth factor (VEGF), platelet-derived growth factor (PDGF), and hypoxia-inducible factor 1α (HIF-1α), are induced by hypoxia. HIF-1α activates the expression of pro-angiogenic factors, leading to hypoxia-induced angiogenesis to provide sufficient oxygen, blood supply, and nutrients for tumor growth. Hypoxia causes a disorganized distribution of the tumor vasculature, which increases the distance between capillaries, resulting in chronic hypoxia and necrosis. This can render tumors resistant to certain types of cancer treatments.

HT combined with RT can help overcome hypoxia in tumors by increasing blood flow to the tumor cells [45]. By increasing the oxygen supply, the activation of HIF-1α is not stimulated to suppress the growth of cancer. Additionally, the heat generated by HT can directly damage tumor cells, making them more vulnerable to the effects of radiation therapy. HT enhances the effect of RT by increasing the oxygen supply to target tumors, namely reoxygenation.

In 1994, Nishimura and Urano observed that HT (43.5 °C, 45 min) prior to RT (5.5–5.6 Gy/min) sensitized the normal tissue response to RT, regardless of whether RT is administered in a single dose or in fractionated doses, with no thermal radiosensitization [46]. Isotransplants of FSa-II and Mca were irradiated under hypoxia or air, and the TCD50 (50% tumor control dose) was assessed. The increasing disparity between the TCD50 values of hypoxia and air without HT shows that considerable reoxygenation occurred during fractionated irradiation. Following fractionated irradiation, the TCD50 (with heat in the air) was less than that of TCD50 (RT alone in the air), indicating that HT did not affect tumor reoxygenation.

Vujaskovic et al. investigated physiological changes induced by RT and HT in spontaneous canine soft tissue sarcomas [47] and effects of local heat on oxygenation, extracellular pH (pHe), and blood flow in spontaneous canine soft tissue sarcomas. Overall, tumor oxygenation improved, tumor perfusion increased reliably with thermal dose (most significant at T50 < 44 °C), and pHe levels decreased at higher T50 values. In addition, the effects of HT on physiological tumor parameters are biphasic. While perfusion and oxygenation are often increased at lower temperatures, vascular damage is prone to occur at higher temperatures, which leads to greater hypoxia.

In 2001, Ressel et al. studied the effects of different treatment modalities on oxygenation [48]. Head and neck squamous cell carcinoma xenografts obtained from humans were treated with various treatment methods and combinations thereof (RT with 5 × 2 or 10 × 2 Gy; HT at 41 °C or 41.8 °C; chemotherapy with ifosfamide or cisplatin). The median value of pO2 increased within the treatment time in all groups except the control group, and the proportion of pO2 below 10 mmHg reduced consistently. The groups with trimodality had the largest difference between the median pO2 values and the fraction of pO2 measurements below 10 mmHg and had the highest rate of CR at day 60.

Thrall et al. reported changes in tumor oxygenation under a fractionated course of combined treatments of RT (25 × 2.25 Gy) and HT (43 °C) using seven canine soft tissue sarcomas [49]. HT improved oxygenation in tumors that had low pretreatment oxygenation and remained throughout the course of fractionated irradiation. This may contribute to the increase in the cell killing effect of RT.

In 2018, Jabbari et al. studied the synergistic effect of HT combined with RT and calcium carbonate nanoparticles (CCNPs) on the proliferation of the human breast cancer cell line, MCF-7 cells [50].

Kim et al. reported that MTH suppresses the RT-induced upregulation of HIF-1 and its target genes by enhancing oxygenation in FSa-II fibrosarcoma tumors in vivo [51]. RT (15 Gy) led to a significant decrease in blood perfusion, an increase in hypoxia, and an upregulation of HIF-1α and VEGF, which promoted revascularization and recurrence. However, MTH (41 °C, 30 min) increased blood perfusion and tumor oxygenation, thereby inhibiting RT-induced HIF-1 and VEGF in tumors, which may result in the increased death of tumor cells and a delay in tumor growth.

Sadeghi et al. revealed that radiation effectiveness is enhanced by the HT-triggered release of the hypoxic cell radiosensitizer pimonidazole (PMZ) from temperature-sensitive liposomes (TSL) [52]. The effectiveness of TSL-PMZ in combination with RT and HT was evaluated in vitro to assess cell survival and DNA damage. Upon heating (42 °C, 5 min), the TSL-PMZ rapidly released substantial levels of PMZ, and the combination of PMZ-loaded TSLs with HT enhanced the effectiveness of RT under hypoxic conditions.

RT and HT can also affect ATP production by damaging cellular DNA, which can disrupt the normal metabolic processes that generate ATP. This can cause cells to have less energy available for repair and make them more vulnerable to RT. In 1987, Sijens et al. investigated the response of murine mammary carcinoma NU-82 cells to RT and HT [53]. The ATP/Pi ratio was found to decrease, particularly at higher temperatures. The changes in phosphodiesters seemed to be correlated linearly with the decrease in ATP, especially when treated with RT (20 Gy) plus HT (44 °C, 15 min). Moreover, heavier doses not only temporarily induce decreases in tumor perfusion but also result in necrosis, as opposed to lower doses.

### 3.2. DNA Damage

Radiation causes the generation of highly reactive free radicals and lethal DNA lesions. Heat stimulates radiation-induced DNA damage by generating more oxidative stress, thus directly damaging tumor cells [54]. These synergistic effects increase the susceptibility of tumor cells to cell death. Several studies have reported that proteins or genes such as RBL1 [55], p53 [56], Rad51 [57], and BRCA2 [58] are involved in DNA repair. Mutations or loss of function of these proteins or genes have been implicated in cancer, and RT + HT has shown an inhibitory effect on cancer proliferation by regulating these factors.

As the tumor suppressor gene *p53* plays a crucial role in cell cycle arrest, apoptosis, and DNA repair inhibition, its mutation might result in ineffective cancer treatment [59]. A study in 2005 showed that the hyperthermic augmentation of tumor growth suppression by irradiation is dependent on the *p53* gene state, using two types of cancer cell lines with different *p53* gene statuses (wtp53, mp53), treated with a combination of X-ray irradiation (2 Gy) and HT (42 °C, 20 min) [59]. Cell growth inhibition was assessed via the Bax and Caspase-3 pathways and the activation of Caspase-3 by PARP and Caspase-3 fragmentation. The hyperthermic augmentation of radiation-induced tumor growth suppression may result in *p53*-dependent apoptosis due to heat-induced inactivation of the cell survival system via control of the cell cycle or promotion of DNA repair.

In 2007, Masunaga et al. demonstrated that MTH inhibits the repair of radiation-induced damage, as measured by the micronuclei (MN) frequency of the total cell and quiescent cell (Q cell) populations in SCC VII tumors in vivo, under high-dose-rate (HDR) or low-dose-rate irradiation immediately followed by MTH (40 °C, 9 h) or the administration of caffeine or wortmannin [60]. MTH effectively reduced the loss in sensitivity in both the total and Q cell populations, thereby reducing the irradiation dose rate. In 2019, they investigated the effects of *p53* status on tumor cells [61]. Human head and neck squamous cell carcinoma cells transfected with mutant TP53 (SAS/mp53) or with neovector (SAS/neo) were injected into nude mice, received HDR immediately, followed by localized MHT (40 °C, 2 h) or caffeine or wortmannin administration. Contrary to the slight recovery of the total or Q tumor cells within SAS/mp3 tumors, SAS/neo tumor cells showed much less sensitivity due to the *p53* recovery from radiation-induced damage. Through two studies, it was concluded that MTH effectively suppressed recovery from radiation-induced damage, as well as wortmannin treatment combined with irradiation.

In 2013, Genet et al. suggested that HT inactivates homologous recombination repair and sensitizes cells to ionizing radiation in a time- and temperature-dependent manner using AG1521 human fibroblast cells and a series of DNA-repair-deficient Chinese hamster cells [62]. At and above 42.5 °C, significant changes in cellular toxicity due to HT were identified. Dissociation and subsequent reformation of Rad51 proteins at DNA double-strand break (DSB) sites in response to HT, which were identified as the major DNA repair proteins, are crucial in HT-induced radiosensitization.

Bergs et al. investigated the effects of HT on genotoxicity and radiosensitization by exposing SW1573 and RKO cells to HT (41 °C, 60 min) prior to irradiation (4 Gy) [63]. Exposure to HT radiosensitized RKO cells by inducing BRCA2 degradation and chromosomal translocation. Chromosomal translocations suggest the genotoxic effects of combined exposure of RT and HT. This was discovered quickly after combined exposure for 1 h but not detectable at 24 h after treatment.

Oorschot et al. suggested that HT and DNA-PKcs inhibitors pre-treatment can inhibit DNA-DSB repair and enhance RT-induced cancer cell cytotoxicity [64]. DSB can repair non-homologous end-joining or recombination. This may affect the efficacy of RT. In this study, suppression of DNA-DSB repair by HT (42 °C, 1 h) and the DNA-PKcs inhibitor NU7441 radiosensitizes human cervical and breast cancer cells and primary human breast cancer sphere cells (BCSCs).

In 2019, Son et al. demonstrated that irradiation (3 Gy) combined with HT (44 °C, 60 min) inhibited the progression of lung cancer via elevated NR4A3 and KLF11 expression, which is critical for enhancing the effectiveness of combined treatment [65]. A549 lung cancer cells exhibited increased levels of NR4A3 and KLF11 after combination therapy, which also induced apoptosis and inhibited cell proliferation by elevating intracellular ROS levels.

Singh et al. observed that, when RT and HT (42 °C, 30 min) were combined, there was additional DNA damage, and no repair was identified 30 min post-irradiation, which may be due to the HT-dependent suppression of DNA repair post-irradiation, as described by previous researchers [66].

Khurshed et al. demonstrated that the addition of HT (42 °C, 60 min) enhanced the efficiency of multimodal therapy with RT, cisplatin, and PARPi in *IDH1*^MUT^, *IDH1*^WT^ HCT116 colon cancer cells, and Hyperthermia1080 chondrosarcoma cancer cells [67]. Isocitrate dehydrogenase 1 (IDH1) is a homodimeric enzyme that catalyzes the conversion of isocitrate to α-ketoglutarate (αKG) via the reduction of NADP+ to NADPH. Mutations in *IDH1* lead to neomorphic IDH activity that converts αKG into the oncometabolite *D*-2 hydroxyglutarate (*D*-2HG), which suppresses the homologous recombination repair system and decreases intracellular reducing power (NADPH), resulting in improved cellular sensitivity to multimodal therapies and tumor progression. The combination of RT and HT resulted in an increase in DSBs and cell death by up to 10-fold in *IDH1*^MUT^ cancer cells compared to *IDH1*^WT^.

### 3.3. Cell Cycle Arrest

Cell cycle arrest is an essential process as it prevents damaged cells from entering mitosis and helps them secure time for investigating their own DNA repair systems [68]. Defects in the G2/M arrest checkpoint allow damaged cells to initiate mitosis and undergo apoptosis, which may enhance the effectiveness of anti-cancer treatments by enhancing their cytotoxicity. HT can induce G2/M phase arrest via the ATM pathway [69]. Radiation also damages DNA and induces cell cycle arrest [70].

In 2002, Yuguchi et al. compared the antiproliferative effects of HT, RT, and combination therapy for esophageal cancer in humans, primarily using cell cycle analysis [71]. It is reported that HT (43.5 °C, 60 min) delayed the progression of cell cycles from the G0/G1 phase to the S-G2/M phase, and RT (5 × 2 Gy) accelerated the inhibition of DNA synthesis. This may result in a strong inhibitory effect on tumors and a high rate of apoptosis, leading to an accumulation of cells in the G2/M phase. The combined treatment enhanced the synergistic effects by inducing apoptosis.

### 3.4. Apoptosis

Apoptosis is triggered by a variety of cellular stressors, including intrinsic stressors (e.g., genomic damage) and extrinsic stressors (e.g., binding to death receptors, heat, radiation, and hypoxia) [72]. Cancer proliferation is enhanced by the loss of balance between pro- and anti-apoptotic proteins, death receptor activation, and the AKT pathway. Several studies have shown that RT + HT can trigger apoptosis by modulating these elements.

#### 3.4.1. Intrinsic Apoptosis

Intrinsic apoptosis is a highly regulated and programmed cellular process that is essential for maintaining tissue homeostasis and eliminating damaged or abnormal cells. A variety of intracellular signals, such as DNA damage, oxidative stress, and the presence of misfolded proteins, activate the intrinsic pathway of apoptosis. The key regulators of intrinsic apoptosis are the Bcl-2 family of proteins, which are classified as either pro-apoptotic or anti-apoptotic based on their function. Pro-apoptotic Bcl-2 proteins, such as Bax and Bak, promote the release of cytochrome C from the mitochondria, which then activates cleavage of caspase enzymes to induce the morphological changes associated with apoptosis (cell shrinkage and fragmentation) [73]. Cancer cells can evade apoptosis by various mechanisms, such as mutations in the Bcl-2 family of proteins, which alter their function and promote cell survival [74]. HT studies demonstrated an induction of the intrinsic apoptosis pathway of cancer cells [75]. HT can also sensitize cancer cells to achieve higher cytotoxicity by RT. Here, we describe studies on this.

Bax plays an essential role in the intrinsic apoptotic pathway. Cancer cells evade apoptosis by upregulating Bcl-2 anti-apoptotic proteins or downregulating pro-apoptotic proteins [76]. Direct activation of the Bax protein stimulates mitochondrial membrane permeabilization and the release of the apoptotic factor cytochrome C, leading to cell death [77].

Liang et al. examined alterations in apoptotic gene expression in human colon cancer cells (HT29) under different treatment modalities and combinations thereof (RT with 10 Gy; HT at 43 °C) [78]. All treatment modalities decreased the expression of p53 and Bcl-2 while increasing the expression of Bax. It was concluded that HT improves the efficacy of RT and chemotherapy against cancers by altering the expression of apoptotic genes.

In 2019, Talaat et al. showed the effectiveness of moderate HT in conjunction with RT for the treatment of hepatocellular carcinoma [79]. In the 40 °C/4 Gy/48 h group, a smaller proportion of viable cells and a high percentage of apoptotic (31%) and necrotic (63%) cells were observed, along with an increase in the expression of pro-apoptotic *Bax* and *FasL* genes, moderate expression of anti-apoptotic *Bcl-2* and *GRP78* genes, and a significant decrease in pro-angiogenic mediators VEGF and PDGF.

#### 3.4.2. Extrinsic Apoptosis and Other Pathways

The extrinsic apoptosis pathway plays a critical role in the immune surveillance of cancer cells. Cancer cells evade extrinsic apoptosis by various mechanisms, such as downregulating death receptors or blocking the downstream signaling cascade, such asAKT pathways. Therapeutic strategies targeting the extrinsic pathway of apoptosis have shown promise in treating cancer, particularly in combination with other therapies. Resistance to extrinsic apoptosis remains a significant challenge in cancer therapy. Applying HT and RT together induces apoptosis (both intrinsic and extrinsic) of cancer cells [79], showing a promising effect as an effective therapeutic strategy.

In 2021, Singh et al. demonstrated that the combination of HT and proton beam radiation (PBRT) might greatly accelerate chordoma cell death by activating the death receptor pathway and apoptosis, which holds promise for the treatment of metastatic chordoma [80]. HT followed by PBRT enhanced cell killing in human chordoma cell lines (U-CH2, Mug–Chor1), exhibiting an RT-dose-dependent decrease in brachyury expression, which could be an indication of chordoma aggression levels and enhanced HSP-70 expression. Overexpression of the brachyury gene is reported to be the most selectively critical gene in chordoma and is targeted as a potential mediator for cancer therapy [81].

AKT is a protein kinase that plays a key role in the PI3K/AKT signaling pathway in the regulation of cell growth, survival, and differentiation. It has been shown that the PI3K/AKT signaling pathway is frequently implicated in the pathogenesis of many cancers and has been validated as a promising therapeutic target [82]. RT and HT have been suggested to inhibit the AKT signaling pathway.

Man et al. showed that HT improves glioma stem-like cell (GSC) radiosensitivity by inhibiting AKT proliferative and pro-survival signaling by identifying the survival kinase AKT as a crucial sensitization factor for GSCs [83]. GSCs treated with HT (42.4 °C, 60 min) prior to irradiation exhibited a decrease in the activation of AKT and proliferation, in contrast to increased AKT activation in GSCs when treated with RT alone. Radiosensitization induced by HT further increased the pharmacological suppression of PI3K.

We have summarized the studies on the effects and mechanisms of RT + HT, including hypoxia, DNA damage, cell cycle, and induction and regulation of apoptosis, in Table 2.

## 4. Immune Response

### 4.1. Cytokines and Antibodies

HT can cause cancer cells to release antigens, which are molecules that help the immune system recognize and target cancer cells more effectively [84]. When combined with RT, HT can enhance immunological responses against cancer by triggering the release of antigens and cytokines (e.g., HSP70 and HMGB1), increasing the activity and number of immune cells, such as natural killer (NK) cells, dendritic cells, and T cells [85]. HSPs can efficiently enhance the immune response [86]. HSP70, a molecular chaperone protein, binds and delivers tumor antigens to antigen-presenting cells when released. Tumor antigens are subsequently presented by dendritic cells (DCs), which stimulate the CD8+ T cell response [87]. In addition, HSP promotes the release of inflammatory cytokines by DCs. Studies focusing on immune responses caused by HT and RT, especially the ones aiming to determine the role of cytokines, were completed in cell models.

In 2009, Schildkopf et al. examined which form of cell death was induced by treating the human colorectal cancer cell line HCT15 with HT (41.5 °C, 60 min) and X-irradiation (5 Gy) [88]. Although the proportion of apoptotic cells did not change, necrosis was the predominant form of cell death following combination treatment. Radioresponsive G2 cell cycle arrest and the release of the danger signal HMGB1 were observed, implying that the combination of RT and HT may contribute to inflammation and immunological activation. In 2010, the same group additionally experimented with human colorectal adenocarcinoma cells with different radiosensitivities using HT (41.5 °C, 60 min) and X-ray irradiation (5 or 10 Gy) [89]. Combinatorial treatment may induce anti-tumor immunity due to the induction of inflammatory necrotic cells and HMGB1 release.

Again, in 2011, Schildkopf et al. demonstrated that RT with HT stimulates the HSP70-dependent maturation of dendritic cells and the release of pro-inflammatory cytokines (IL-8 and IL-12) by dendritic cells and macrophages, in vitro and in vivo, using HCT15 and SW480 [90]. The same HT condition was used, being combined with 2, 5, or 10 Gy X-rays. Combination therapy could boost the expression of surface HSP70, as well as significantly upregulate the co-stimulatory protein CD80 and chemokine receptor CCR7 on DC in a manner dependent on extracellular HSP70.

Wang et al. investigated the effects of magnetic induction HT (MIH) on a 4T1 mouse model of metastatic breast cancer [91]. When treated with MIH and RT (6-MV X-rays), there was a substantial decrease in tumor volume and lung metastasis, improvement in survival and Bax expression, and greater CD4 + T cell percentage and CD4 + /CD8+ cell ratio than with RT or MIH alone. MIH promotes the anti-tumor effect of RT via Bax-mediated cell death, enhances the immunity of cells undergoing RT, and suppresses the increase in matrix metalloproteinase-9 (MMP-9) expression induced by RT.

In 2016, Werthmöller et al. discovered that HT (41.5 °C, 60 min) combined with RT (2 Gy) stimulates the immunogenic potential more than RT alone, which may result in anti-tumor immunity [92]. To test immunity, radioresistant melanoma B16-F10 cells were derived from C57/BL6 mice. The combination of RT and HT enhanced apoptosis, necrosis, the release of HMGB1 and Hsp70, and the infiltration of CD8 + T cells, DCs, and NK cells.

Mahmood et al. discovered that HT (42.5 °C, 30 min) combined with RT (8 Gy) is efficient in reducing tumor volume and increased cytotoxic CD8a + T cells in C57BL/6 mice injected with Panc02 cells [93]. In addition, they discovered that the combination of RT, HT, and immunotherapy increased the number of CD4 + T cells.

### 4.2. NK Cells

RT and HT can stimulate the immune system, including NK cell activation [94]. RT simultaneously causes the release of cytokines, attracts NK cells, and induces NK cells to kill cancer cells [95]. NK cells kill infected or cancerous cells by rupturing the membrane of the target cell and releasing cytokines that activate other immune cells [96]. Because of the release of cytotoxic granules by NK cells, cancer cells undergo apoptosis.

In 2016, Hietanen et al. studied NK cell cytotoxicity and its recovery after HT (31–45 °C, 0–180 min) or HT with RT [97]. From 31 °C to 37 °C, cytotoxicity remained unchanged but decreased as the temperature rose; at 43 °C, cytotoxicity was almost zero. HT (42 °C, 30 min) with RT (20 Gy) reduced cytotoxicity more than HT alone. The ratio of normal to tumor cells killed increased as the temperature increased. NK cell cytotoxicity was unaffected by the sequence or time interval. In 2018, they discovered that HT with RT decreased the ATP levels in NK cells, which determined the probability of cell death [98]. As HT duration increased, the ATP level of NK cells was decreased by RT after HT compared to that after RT. IL-2 restored cell viability and cytotoxicity after exposure to RT and HT.

Finkel et al. discovered immunogenic potential, especially the function of NK cells, in B16 melanoma cells treated with RT (15 Gy) and HT (41.5 °C, 60 min) [99]. RT plus HT enhances apoptosis, necrosis, HMGB1 release, and NK cell count in vitro. HT and RT substantially increased the number of infiltrating B cells (CD3-CD19+), NK cells (CD3-NK1.1+, CD27 + CD11b−), and T cell (CD3+) subpopulations (CD8 + CD4−, CD8 + CD4+) in vivo. A prolonged reduction in NK cells two days after HT and RT enhanced tumor growth. In contrast, the depletion of a single NK cell prior to RT and HT dramatically slowed tumor growth.

### 4.3. Immune Checkpoint Molecules (ICMs)

ICMs regulate the immune responses in cancer cells. Checkpoint inhibitors bind to ICMs to prevent cancer cells from evading the immune system, thereby allowing the immune system to recognize and eliminate them [100]. HT enhances ICMs expression, thereby making cancer cells more accessible to the immune system and more susceptible to immune attack [101]. Radiation treatment can damage the DNA of cancer cells, prompting them to produce more molecules that bind to ICMs and activate the immune system [102].

In 2020, Hader et al. discovered that cell death and immune-modulatory capabilities increased with increasing HT or HT (39, 41, and 44 °C, 60 min) + RT with normofractionation (2 Gy) or hypofractionation (5 Gy) in MCF-7 and MDA-MB-231 human breast cancer cells [103]. HT was performed using either warm water (CH) or microwave heating (MH). The combination of HT with RT enhanced apoptosis and necrosis, and the expression of ICMs, especially hypofractionation (5 Gy), was more effective. RT with HT by MH was more effective in cell death at 41 and 44 °C in MCF-7 cells and at 39, 41, and 44 °C in MDA-MB-231 cells. The expression of immune suppressive ICMs (PD-L1, PD-L2, and HVEM), immune stimulatory ICMs (CD137-L, Ox40-L, CD27-L, and ICOS-L), and EGFR in breast cancer cells was enhanced by RT with HT, consequently enhancing anti-tumor responses. In 2021, they discovered that immunosuppressive ICMs were affected at all temperatures in murine B16 melanoma cells but not at temperatures above 44 °C in human cell lines [104]. HT by 2.45 GHz at 44 °C and RT (5 Gy) produced the highest rate of cell death regardless of HT duration.

In 2022, Sengedorj et al. discovered that cell death and immune phenotype of human MDA-MB-231 and MCF-7 breast cancer cells were not affected by the sequence of radiation and HT but rather by the combination of RT (5 Gy) [105]. Following RT and HT, apoptosis was the main method of cell death in MCF-7 cells, while both apoptosis and necrosis were observed in MDA-MB-231 cells. RT and HT increased the ICM expression (PD-L1, PD-L2, and HVEM) compared with RT alone.

Inhibitory ICMs (PD-L1, PD-L2, and HVEM) were considerably enhanced in MCF-7 breast cancer cells following 120 h of RT with HT (39, 41, 44 °C) treatment. MDA-MB-231 breast cancer cells increased PD-L1 expression early (24, 48 h) after exposure to RT and HT at 44 °C. RT with HT of 41 or 44 °C increased PD-L2 expression at all time points, although HVEM expression was only slightly influenced at earlier time points. In addition, OX40-L, an immunostimulatory ICM, was considerably increased, especially 120 h after RT and HT.

Stoll et al. showed the effect of RT, HT, and the combination of the two (RHT) on cell death mechanisms, the expression of ICM, and the release of the danger signal HSP70 in two human glioblastoma cell lines (U87 and U251) [101]. Particularly in U251 cells, combination therapy induced a significant increase in both apoptosis and necrosis. Moreover, it increased the release of HSP70, mainly after HT (44 °C) only or in combination with RT. A significant increase in immune suppressive (PD-L1, PD-L2, HVEM) and immune stimulatory (ICOS-L, CD137-L, and Ox40-L) ICM was found mostly in U87 cells, particularly after RHT at 41 °C.

Kim et al. studied C3H mice with FSa-II fibrosarcoma subjected to RT (15 Gy) and mild-HT (41.0 °C, 30 min) [106]. The combination of RT and HT disrupted the expression of HIF-1, VEGF, and PD-L1 and dramatically inhibited tumor growth. We have summarized the research on the effects and mechanisms of HT and RT on the immune response in Table 3.

## 5. Discussion

We reviewed 61 experimental studies of combined RT and HT therapy. According to the research era and subject matter, these studies can be separated into three distinct phases as follows:

In phase 1, neither the type nor the mechanism of cell death was identified; only the involvement of hypoxia was revealed.

In phase 2, researchers focused on hypoxia-induced DNA damage and cell cycle arrest.

In phase 3, research was conducted to uncover the mechanism of apoptosis and optimize the RT + HT technique.

In the following discussion, the characteristics of each phase of research are reviewed, and, based on this, the experimental research results of RT + HT are described, along with the research trends of RT + HT and future application approaches.

### 5.1. Discovery of Effects by the Combination of RT + HT (Phase 1)

Through in vivo studies, researchers initially discovered the effects of the combination therapy of RT and HT on tumors. A series of studies focused on cell death, tumor volume, or growth rate but did not explain the mechanism in detail. There were seventeen papers studied in vivo: dog, three; rabbit, one; rat, two; mouse, eleven. Researchers have experimented with various tumor types, including oral or external nasal fibrosarcoma (one), murine sarcoma (two), murine mammary carcinoma (three), primary malignant melanoma (one), mast cell sarcoma (one), skin papilloma (one), rhabdomyosarcoma (one), murine fibrosarcoma (two), murine melanoma (two), human-derived head and neck squamous cell carcinoma (one), and lung cancer (one). A study in 2015 experimented with a non-tumor model [32].

In 2006, the first in vitro study was initiated, which investigated the best exposure period and dose of RT with HT on human cancer cells. Since then, in vitro studies have been more frequent than in vivo studies, as discussed in ten and three papers, respectively. Since this effect has been demonstrated by in vivo studies, several in vitro studies have been carried out to maximize the effect of combined therapy. One of nine studies focused on the best exposure time and dose of HT or RT [38], four studies focused on combination with other therapies [31,34,35,42], two studies focused on spheroidal culture [36,39], and two studies focused on the effect [33,37]. Researchers have attempted to add a variety of medications, design a new research model using the latest technology (e.g., 3D tumor spheroids), and modify the concentration of the current treatment. In other words, optimization was sought to enhance the efficacy, followed by studies to minimize toxicity.

### 5.2. Exploration of the Physiological Changes Following the Demonstration of Effects (Phase 2)

Twenty-four studies were included in this classification. These studies focused on hypoxia (eight), DNA damage (nine), cell cycle arrest (three), induction and regulation of apoptosis (four; Bax (two), Brachyury (one), AKT (one)). Proteins and transcription factors associated with apoptosis, such as Bax and AKT, were also identified. Recent studies have appeared to be more interested in apoptosis than necrosis, as opposed to earlier investigations that revealed necrosis. Additionally, it is noticeable that there have been several studies that have concentrated on identifying optimal conditions to avoid necrosis.

RT and HT can directly damage cancer cells and cause DNA damage, which disrupts ATP production and renders cells more vulnerable. We reviewed eight studies that demonstrated the effects of combined therapy with oxygenation. A study in 1987 demonstrated that combination therapy decreased the ATP/Pi ratio [53]. Four studies showed that combination therapy promoted reoxygenation but did not specify the type of cell death that occurred. One study reported a decrease in TCD50 (50% tumor control dose) [46], one an increase in tumor perfusion [47], one an increase in pO2 [48], and another without any mechanism [49]. Three papers discussed the molecular mechanisms in greater depth. One study reported an increase in ROS production and the expression of Caspase-3 and -9 [50], while another reported the suppression of RT-induced upregulation of HIF-1α, VEGF, and CA9 [51]. These two studies demonstrated the occurrence of apoptosis. Another study showed that the addition of radiosensitizers enhances cell death and DNA damage [52], in addition to two additional interventions other than RT and HT, such as CCNPs [50] and TSL-PMZ [52].

HT can hinder recovery from radiation-induced damage and normal cell function. This can lead to the accumulation of DNA damage, resulting in cell cycle arrest in the G2/M phase and increased radiosensitivity. In addition, RT can directly cause DNA damage, which can trigger checkpoint pathways that halt the cell cycle. DNA damage and cell cycle arrest are discussed in sections 9 and 1, respectively. Five of nine mentioned apoptosis as the type of cell death, two mentioned necrosis, and the rest did not. In 2002, a study showed chromosomal and DNA abnormalities, cell cycle delay by HT, and inhibition of DNA synthesis by RT [71]. Researchers have demonstrated p53-dependent inhibition of tumor growth via Bax and apoptosis-related proteins (e.g., PARP, Caspase-3) [59,61]. According to a 2007 study, HT effectively suppressed the loss in sensitivity, as indicated by the MN frequency [60]. Three studies indicated that HT impairs homologous recombination repair. One study addressed the dissociation and reformation of Rad51 proteins as crucial in radiosensitization by HT [62], while another reported chromosomal abnormalities and transient BRCA2 degradation [63]. Another study found that mutations in IDH1 lead to the accumulation of the oncometabolite *D*-2HG, which suppresses homologous recombination repair and lowers NADPH [67]. According to a study in 2016, inhibition of DSB repair promotes cell cytotoxicity, cell cycle arrest, and Caspase-3 activation [64]. A study in 2019 found that the upregulation of KLF11 and NR4A3 is associated with the induction of apoptosis and ROS generation and is critical for slowing the progression of lung cancer [65]. Overall, hypoxia and reoxygenation induced by RT + HT can trigger structural damage and DNA lesions, leading to cell cycle arrest or cell death. When HT is paired with RT, it exacerbates radiation-induced oxidative stress and suppresses the recovery of radiation-induced DNA damage. HT can improve the efficacy of tumor treatment.

An imbalance between pro- and anti-apoptotic proteins usually causes apoptosis in cancer cells. The pro-apoptotic gene, Bax, is involved in the intrinsic apoptotic pathway. All four studies referenced apoptosis and one study published in 2019 indicated necrosis at the same time. Two studies demonstrated changes in the expression of apoptosis-related genes, including p53, Bcl-2, and Bax [78,79]. One study demonstrated the suppression of pro-angiogenic factors, such as VEGF and PDGF [79]. The PI3K-AKT pathway is often aberrantly regulated in cancer cells. AKT is a protein kinase that plays a key role in regulating cell growth, survival, and differentiation. HT and RT may suppress the AKT signaling pathway, rendering cells more vulnerable to harm and causing DNA damage. Research published in 2015 showed that HT improves radiosensitivity by reversing the stimulation of AKT signaling and its downstream gene expression produced by RT, which is further enhanced by the suppression of PI3K [83]. A study in 2021 revealed an increase in HSP-70 expression, downregulation of the brachyury gene, and activation of the death receptor and apoptotic pathway [80].

### 5.3. Immune Response (Phase 3)

By combining RT and HT therapy, we identified 14 papers that applied the mechanisms of the anti-tumor immune response. The detailed mechanisms studied were HMBG1 (two studies), pro-inflammatory cytokine (one), CD8 + T cells (three), NK cells (three), and ICMs (five).

Recently, immunotherapy has been suggested as a new anti-tumor therapeutic target. Immunotherapy offers several advantages over conventional cancer therapy. It provides a targeted approach, can have long-lasting outcomes, and can enhance patients’ quality of life [107].

Recent research has indicated that RT with HT improves the immune response compared to RT alone. By heating the tumor, tumor cells can be forced to release antigens, thereby stimulating an immune response. Two studies referred to necrosis as the prevalent form of cell death and discussed the release of the danger signal HMGB1 and its contribution to immune activation [88,89]. HT and RT induce immune responses dependent on HSP70, including the upregulation of dendritic cell markers (CD80 and CCR7) and secretion of pro-inflammatory cytokines (IL-8 and IL-12) [92].

RT combined with HT can also increase the activity of immune cells, such as CD8+ T cells, CD4+ T cells, dendritic cells, and NK cells, resulting in tumor death. CD8+ and CD4+ T cells play a vital role in the killing of tumor cells, both in terms of their direct anti-tumor activity and their support for the immune response. Three papers studied T cells, one study focused on the CD4+ T cell percentage and CD4+/CD8+ cell ratio [91], one on the infiltration of CD8 + T cells [92], and one on cytotoxic CD8a+ T cells [93].

By combining RT with HT, NK cell activation, a defense mechanism against cancer cells, is significantly enhanced. NK cells target cancer cells to suppress proliferation by producing cytokines and inducing apoptosis. Three papers studied NK cells: one focused on NK cell cytotoxicity [97], one on NK cell count [99], and one on the ATP level of NK cells [98]. In addition, one study found that NK cells have a dual- and time-dependent effect on the efficiency of the anti-tumor immune response [99].

ICMs have recently become the subject of many studies on immune responses. ICMs, which are present on the surface of cancer cells, impair the ability of cancer cells to avoid the immune system. ICMs include immunostimulatory ICMs (CD137-L, Ox40-L, CD27-L, and ICOS-L) and immunosuppressive ICMs (PD-L1, PD-L2, and HVEM). The combination of RT and HT can enhance the expression of ICMs and the production of immune cells that are capable of targeting cancer cells [101,103,104,105,106]. The interaction between RT with HT and ICMs is complicated; thus, new research has been conducted from 2020 to the present that looks at altering the time, sequence, cell line, and frequency [104,105]. Table 4 shows a summary of the studies on the combination therapy of RT + HT, published from 1984 to the present, by theme and period.

Overall, the combination therapy of RT and HT has a synergistic effect on cancer cells, rendering them more susceptible to cell death and ultimately improving treatment outcomes. Initial research was primarily concerned with identifying whether the two treatments have synergistic effects that enhance radiosensitivity. They did not reveal the exact mechanism but only mentioned the involvement of hypoxia. Based on the success of combination therapy established in earlier trials, the discovery of precise mechanisms began in phase 2. The goal of the studies in phase 3 has been to investigate the conditions for optimizing therapeutic efficacy and uncover mechanisms more precisely. Several studies in the third period have suggested that this mechanism is closely linked to the regulation of the immune response.

## 6. Conclusions

Cancer poses a threat to human health and wellbeing. Despite the biphasic nature of HT, it has a synergistic effect when paired with RT, making it a viable option for treating cancer. Table 4 summarizes the efficacy of RT + HT combination treatment and the trend of RT + HT research. RT was performed mostly within the range of 2–10 Gy with certain exception studies, which used up to 100 Gy. HT was relatively more consistent; 41–44 °C for a period not exceeding 60 min was the standard condition. Certain studies used lower (39–40 °C) or higher temperatures (45–47 °C) and a longer time period (90 min). Several limitations remain of such combination therapy. The purpose of introducing HT in combination with RT is mainly to sensitize tumor cells to RT without affecting the normal tissues. This is supported by extensive evidence. However, RT itself is not applied tumor-specifically; thus, we cannot rule out the effect on normal tissues. Further detailed investigation is necessary to clearly determine how RT and HT simultaneously affect non-malignant cells. Such efforts will increase the potential application of the RT + HT combined therapy to the clinical field. Recently, conventional cancer therapies, such as chemotherapy, RT, and surgical resection, have been gradually replaced by more advanced immunotherapy. The combination of RT and HT activates the immune system. This may be the most fundamental therapeutic strategy for the future. Although further studies are required, RT and HT may provide an important advancement to immunotherapy since they can impact the immune system of an individual. When the safety profile of RT and HT combination therapy is fully demonstrated, its potential will expand considering the recent cancer therapy trend involving the immune system. Overall, RT and HT are effective cancer treatments that complement conventional treatments while enhancing immunotherapy, which has the potential to improve cancer treatment outcomes in the future.

## Figures and Tables

**Table 1 antioxidants-12-00924-t001:** Effect of RT + HT on tumor cells (unknown mechanism).

RT	HT	Cell Line and Observation Model	Classification of the Molecular Mechanism	Ref.
32–48 Gy	50 °C, 30 s 43 °C, 30 min	Oral or external nasal fibrosarcoma, ten dogs/in vivo	Necrosis	[17]
-	-	Murine sarcoma, sarcoma 180, C3H mice/in vivo		[18]
51.08 Gy	43 °C, 10 min	Murine mammary carcinoma, FM3A, C3H mice/in vivo		[19]
36.8 Gy (8 × 4.6 Gy)	43 °C, 20 min	Primary malignant melanoma, 43 dogs/in vivo		[20]
45.5 Gy (13 × 3.5 Gy)	44 °C, 30 min	Mast cell sarcoma, a dog/in vivo	Necrosis	[21]
17 Gy (2 × 8.5 Gy)	43.5 °C, 30 min	Breast carcinoma, Tx; the sarcoma 37, S37, BALB/C male mice/in vivo	Necrosis	[22]
10 Gy	46 °C, 60 min	Shope-virus-induced skin papilloma, VX-2, rabbits/in vivo	Necrosis	[23]
30 Gy	44 °C, 30 min	Rhabdomyosarcoma, R-1, Wag/Rij female rats/in vivo		[24]
7.2 Gy/min (total dose not reported)	45.5 °C, 10 min; followed by 41.5 °C, 60 min	Murine fibrosarcoma, FSa-II, C3Hf/Sed mice/in vivo		[25]
5.5–5.6 Gy/min (86.2–101.7 Gy for five days)	43.5 °C, 45 min	Murine fibrosarcoma, FSa-II, C3Hf/Sed mice/in vivo		[26]
10 Gy	43 °C, 30 min	Murine melanoma, B16F1, C57BL mice/in vivo	Apoptosis	[27]
10 Gy (5 × 2 Gy)	41.8 °C, 60 min	Human-derived head and neck squamous cell carcinoma, athymic nude mice/in vivo		[28]
10 Gy	43 °C, 30 min	Murine sarcoma, Sarcoma 180 (S180), Balb/c mice/in vivo	Necrosis	[29]
30 Gy	43 °C, 30 min	Murine melanoma, B16F1, C57BL mice/in vivo		[30]
0.5 Gy/min (total dose not reported)	41 and 43 °C, 30 min	Human origin cervical carcinoma, SIHA; non-small-cell squamous lung carcinoma, SW-1573; colon cancer, RKO; rodents cell line V79, R1 and RUC/in vitro		[31]
9 Gy	43 °C, 60 min	396 Wistar rats / in vivo		[32]
36 Gy	43 °C, 60 min	Human glioblastoma, U-87MG/in vitro		[33]
2.5 Gy/min (total dose not reported)	40 °C, 60 min	Lymphoma, EL4, C57BL/6J mouse/in vitro	Apoptosis	[34]
2 Gy	42 °C, 60 min	Cervical cancer, SiHa and HeLa/in vitro		[35]
6 Gy	43 °C, 60 and 90 min	Human prostate cancer stem cells (CSCs), DU145/in vitro		[36]
5 Gy	42 °C, 30 min	Rat gliosarcoma, Gs-9L; non-cancerous tissue of canine kidney, MDCK; human-derived breast cancer, MCF-7/in vitro	Apoptosis	[37]
10 Gy (2 × 5 Gy)	42 °C, 30 min	Lung cancer, A549 and NCI-H1299, BALB/c nude mice/in vitro, in vivo	Apoptosis	[38]
2 and 5 Gy	47 °C, 0–780 CEM43	Human colon cancer, HCT116; oral squamous carcinoma, CAL27/in vitro	Apoptosis, Necrosis	[39]
10 Gy	45 °C, 30 min	Human head and neck cancer, FaDu; human glioblastoma, T98G; human prostate cancer, PC-3/in vitro		[42]
44 and 46 Gy	41.5 °C, 60 min	Mammary carcinoma cell line, C3H, CDF1 mice/in vivo		[43]

**Table 2 antioxidants-12-00924-t002:** Oxygenation, DNA damage and cell cycle arrest, induction, and regulation of apoptosis.

RT	HT	Cell Line and Observation Model	Classification of the Molecular Mechanism	Molecular Mechanism	Ref.
20 Gy	44 °C, 15 min	Murine mammary carcinoma, NU-82, DBA-2 mouse/in vivo	Necrosis/ ATP depletion	Pi ↑, ATP and phosphodiesters ↓	[53]
102.8 Gy (FSa-II); 40.4 Gy (MCa)	43.5 °C, 45 min	Spontaneous murine fibrosarcoma, FSa-II; mammary carcinoma, MCa, C3Hf/Sed mice/in vivo	Oxygenation	-	[46]
56.25 Gy (25 × 2.25 Gy)	44 °C	Spontaneous canine soft tissue sarcomas, 13 dogs/in vivo	Oxygenation	pO2 ↑, tumor perfusion ↑, hypoxic fraction ↓, pHe ↓	[47]
10 Gy (5 × 2 Gy)	41.8 °C, 60 min	Human-derived head and neck squamous carcinoma, athymic nude (nu-nu) mice/in vivo	Oxygenation	pO2 ↑	[48]
10 Gy (5 × 2 Gy)	43.5 °C, 60 min	Human esophageal carcinoma SGF-3, −4, −5, −7, −8, and −9/in vitro	Cell cycle arrest	Chromosomal aberrations, G2/M phase accumulation	[71]
2 Gy	42 °C, 20 min	Cancer cell lines carrying a different p53 gene status (wt p53 and m p53)/in vitro	Apoptosis/ DNA damage	p53–dependent apoptosis, Bax and Caspase-3 pathways	[59]
56.25 Gy (25 × 2.25 Gy)	43 °C, CEM43°CT90 = 10 and 40 min	Canine soft tissue sarcomas/in vitro	Oxygenation	-	[49]
10 Gy	43 °C, 60 min	Human colon cancer, HT29, nude mice/ in vivo	Apoptosis/ Bax	p53 and Bcl-2 ↓, Bax ↑	[78]
2.75 Gy/min (total dose not reported)	40 °C, 9 h	Squamous cell carcinoma, SCC VII, C3H/He mice/in vivo	DNA damage	Change in MN frequency	[60]
1 Gy	42.5 °C, 60 min	Chinese hamster ovary cells, CHO WT (CHO 10B2); normal human fibroblast cell line, AG1521; DNA repair deficient CHO mutants, V3 (DNA-PKcs), irs1SF (XRCC3), KO40 (FancG), 51D1 (Rad51D), and xrs5 (Ku80); V79 mutants irs1 (XRCC2) and irs3 (Rad51C)/in vitro	DNA damage	Chromosomal aberrations, Rad51 activity at DSBs	[62]
2 Gy	42.4 °C, 60 min	GSC, patient specimens 3691 and 387, athymic nude mice/in vitro, in vivo	Apoptosis/ AKT	DNA repair ↓, AKT Signaling ↓	[83]
4 Gy	41 °C, 60 min	Human lung carcinoma, SW-1573; human colorectal carcinoma, RKO/in vitro	Apoptosis/ DNA damage	Chromosomal aberrations and translocation, BRCA2 degradation, homologous recombination pathway ↓	[63]
4 Gy	42 °C, 60 min	Human cervical cancer, HeLa and SiHa; human breast cancer, MCF7, and T47D; primary human breast cancer, BCSC, athymic mice/in vivo (SiHa), in vitro	Apoptosis/ DNA damage	DNA-DSB repair ↓, G2/M phase arrest, Caspase-3 activity ↑	[64]
2 Gy	41 °C, 60 min	Human breast adenocarcinoma, MCF-7/in vitro	Apoptosis/ Oxygenation	ROS ↑, Caspase-3 and –9 ↑	[50]
15 Gy	41 °C, 30 min	Fibrosarcoma, FSa-II, C3H mice/in vivo	Apoptosis/ Oxygenation	HIF-1a and VEGF ↓	[51]
4 Gy	40 °C, 48 h	Liver cancer, HepG2/in vitro	Apoptosis, Necrosis/ Bax	Bax and FasL ↑, VEGF and PDGF ↓	[79]
8.5, 14, and 21 Gy (LDR); 4, 8.5, 14.5, 20, and 24 Gy (HDR)	40 °C, 2 h	Human head and neck squamous cell carcinoma, SAS, nude mice/in vivo	Apoptosis/ DNA damage	Tumor sensitivity ↓, p53-dependent recovery	[61]
4 Gy	42 °C, 5 min	Human hypopharyngeal carcinoma, FaDu/in vitro	Oxygenation	histone γH2AX phosphorylation ↑	[52]
3 Gy	44 °C, 60 min	Human non-small-cell lung cancer, A549 and NCI-H292/in vitro, in vivo	Apoptosis/ DNA damage, HSP70	KLF11 and NR4A3↑, intracellular ROS ↑	[65]
2 Gy	42 °C, 30 min	DNA extracts/in vitro	DNA damage	-	[66]
-	-	Human chordoma, U-CH2 and Mug-chor1/in vitro	Apoptosis/ gene expression, HSP70	Brachyury ↓, death receptor activation	[80]
2 Gy	42 °C, 60 min	Colon cancer, IDH1^MUT^ and IDH1^WT^ HCT116; chondrosarcoma, Hyperthermia1080 cells/in vitro	DNA damage	NADPH↓, homologous recombination repair ↓	[67]

**Table 3 antioxidants-12-00924-t003:** RT + HT induces the immune response in tumor cells.

RT	HT	Cell Line and Observation Model	Classification of the Molecular Mechanism	Molecular Mechanism	Ref.
5 Gy	41.5 °C, 60 min	Human colorectal adenocarcinoma, HCT15/in vitro	Necrosis/ Immune response	HMGB1 release, G2/M phase arrest	[88]
5 or 10 Gy	41.5 °C, 60 min	Human colorectal adenocarcinoma, SW480, and HCT 15/in vitro	Necrosis/ Immune response	HMGB1 release, G2/M phase arrest	[89]
2, 5, and 10 Gy	41.5 °C, 60 min	Human colorectal tumor, HCT15, and SW480; mouse colon carcinoma tumor, CT26.WT (CRL-2638)/ in vitro, in vivo	Immune response, HSP70	CD80 and CCR7 ↑, phagocytosis of macrophages and DCs ↑, IL-8, and IL-12 ↑	[90]
20 Gy (2 × 10 Gy)	45 °C, 3 min	Human murine breast cancer, 4T1, BALB/C mice/in vivo	Apoptosis/ Immune response	CD4+ T cell and CD4+/CD8+ cell ratio ↑, TNF-α, IFN-γ, and IL-2 ↑, Bax ↑, MMP-9 ↓	[91]
2 Gy	41.5 °C, 60 min	Mouse melanoma, B16-F10, C57/BL6 mice/ in vitro, in vivo	Apoptosis, Necrosis/ Immune response, HSP70	HMGB1 ↑, infiltration of CD8 + T cells, DCs, and NK cells ↑	[92]
20 Gy	42 °C, 30 min	Erythroleukemia, K-562/in vitro	Immune response	NK cytotoxicity ↓	[97]
15 Gy	41.5 °C, 60 min	Mouse melanoma, B16-F10, C57BL/6 mice/ in vivo, in vitro	Apoptosis, Necrosis/ Immune response	HMGB1 release, NK cell, B cell, and T cell count ↑	[99]
20 Gy	42 °C, 0–180 min	NK cell; Erythroleukemia, K-562/in vitro	Immune response	ATP level of NK cell ↓	[98]
8 Gy	42.5 °C, 30 min	Murine syngeneic Panc02, Panc02, C57BL/6 mice/in vivo	Immune response	CD8a+ and CD4+ T cells ↑	[93]
10 Gy (5 × 2 and 2 × 5 Gy)	39, 41, 44 °C, 60 min	Human breast cancer, MCF-7, and MDA-MB-231/in vitro	Apoptosis, Necrosis/ ICM, HSP70	PD-L1, PD-L2, HVEM ↑, CD137-L, OX40-L, CD27-L, ICOS-L ↑, EGFR ↑	[103]
10 Gy (5 × 2 and 2 × 5 Gy)	39, 41, 44 °C, 60 min	Murine melanoma, B16; human breast cancer, MCF-7, and MDA-MB-231/in vitro	Apoptosis, Necrosis/ ICM	PD-L1, PD-L2, HVEM, and Gal-9 ↑	[104]
10 Gy (5 × 2 Gy)	41 and 44 °C, 1 h	Human glioblastoma, U87 and U251/in vitro	Apoptosis, Necrosis/ ICM, HSP70	PD-L1, PD-L2, HVEM ↑, ICOS-L, CD137-L, and Ox40-L ICMs ↑	[101]
10 Gy (2 × 5 Gy)	39, 41, 44 °C, 60 min	Human MCF-7 and MDA-MB-231 breast cancer cells/in vitro	Apoptosis, Necrosis/ICMs	PD-L1, PD-L2, and HVEM ↑	[105]
15 Gy	41.0 °C, 30 min	Fibrosarcoma, FSa-II, C3H mice/in vitro	Apoptosis/ ICM, Hypoxia	PD-L1 ↓, VEGF ↓, HIF-1α ↓	[106]

**Table 4 antioxidants-12-00924-t004:** Number of theses by period.

Theme (Table No.)	Period 1 (1984 ~2001)	Period 2 (2002 ~2010)	Period 3 (2011 ~2022)	Total
Unknown mechanism (Table 1)	11	3	11	25
Hypoxia; DNA damage and Cell cycle arrest; Induction and regulation of apoptosis (Table 2)	4	5	13	22
Immune response (Table 3)	0	2	12	14
Total	15	10	36	61

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
