# Peer review of "Combination Therapy of Radiation and Hyperthermia, Focusing on the Synergistic Anti-Cancer Effects and Research Trends"

_antioxidants, 2023, doi:10.3390/antiox12040924_

Round 1

Reviewer 1 Report

The manuscript is a review of existing literature on the combination treatments of hyperthermia and radiation therapy. While the subject is interesting, this method is not currently widely used in clinic, with very little uptake in a few centres and for very few indications. In current form, the manuscript is not appropriate for publication, and subject to revisions.

The review is well written, but it lack a critical approach of all the work presented here, and a critical synthesis of all previous literature. The field needs a rigorous review, highlighting any temperature response of hyperthermia and radiotherapy, and highlight any dose enhancement factor collated from all previous in vivo and in vitro studies. This review is a description of previous literature, with minor highlights on suspected mechanisms of action previously discussed.

Several specific comments below:

1.       The authors state ‘Normal tissue is rarely affected by temperatures below 45 degrees’, and reference which just state this with not further explanation, and does not bring any proof to this statement. The normal tissue toxicity is not clearly discussed, and investigated in the hyperthermia work, which is part of the reason for the low uptake past preclinical testing.

2.       Page 2, lines 108-109 the authors talk about a cure rate. What is a cure rate? Define cure? How long were the mice alive, after ‘cure’? what was the tumour growth delay for the discussed method?

3.       Throughout the manuscript, author report the radiation dose inconsistently, sometimes, the dose per fraction ( and no total number of fractions described) or just the dose rate. Please clarify on the total dose used in these studies.

4.       Line 117, the authors state that there was ‘No visible thermal injury’? What does that mean? Is there a scale to mark these injuries? Please clarify.

5.       Page 4, lines 153-154 the authors seem to confuse the dose and dose-rate and declare the dose-rate inside

6.       Page 9, first paragraph, lines 340-355 the last part of the paragraph ends in a very confusing way. Please rephrase and clarify.

7.       The immune response (section 4.1) and cytokine effects are largely discussed in vitro, please state that. Also, these effects are known to be largely radiation dose dependent. Please provide dose information on the cytokine response.

8.       Page 17, lines 612- the authors state that “HT can hinder recovery from radiation-induced damage and normal cell function.”  Is this a potential mechanism for radiation induced toxicity? It sure sounds like it. As I previously said, in this field, all normal tissue effects are very poorly reviewed and detailed, which is what the field needs.

9.       Are the authors suggesting any largely used temperature, time interval and radiation dose used for a successful hyperthermia and radiotherapy combination? How would this change for different cancer indications?

10.   Are there any weaknesses in the field that need to be addressed to enable clinical translation of these findings?

11.   Do the authors recommend anything to improve the future research output in the field?   

Reviewer 2 Report

Dear authors,

After reviewing your publication, I have several concerns and suggestion:

1. Abstract

Radiation therapy (RT) is an essential component of cancer treatment. - "essential" is not true. RT is not always used.

2. in the abstract or at least in the introduction, there is no information that the data analysis concerns research on humans, animals and cell lines.

3.27: "According to statistics from the American Cancer Society, there will be approximately" the authors write "approximately" but immediately present a specific number of patients.

4. In the introduction, both HT and RT, the effects of both methods separately are very superficially written. They should be more developed based on the literature.

5. Information/data concerning experiments conducted on animals and those concerning experiments conducted on cell lines should be separated into individual subsections.

6. Sections are rather superficially written (information is missing. (Apoptosis is triggered by a variety of cellular stressors, including intrinsic stressors (e.g., genomic damage) and extrinsic stressors (e.g., binding to death receptors, heat, radiation, and hypoxia). information can be summarized by adding 3-4 sentences.The sections on apoptosis are based on a small number of citations, from 1 to 4, but the section on Gene expression contains only two citations.

Chapter 4. Immune response is best written. Authors should edit the publication on the model of chapter 4.

Round 2

Reviewer 2 Report

Dear authors,

Thank you for making the corrections.

Now the publication looks much better.

Best!